# Medication Dispensing by Pharmacy Technicians Improves Efficiency and Patient Safety at a Geriatric Ward at a Danish Hospital: A Pilot Study

**DOI:** 10.3390/pharmacy11030082

**Published:** 2023-05-08

**Authors:** Lene Juel Kjeldsen, Maja Schlünsen, Annette Meijers, Steffan Hansen, Camilla Christensen, Tanja Bender, Barbara Ratajczyk

**Affiliations:** 1The Hospital Pharmacy, University Hospital Sønderjylland, 6200 Aabenraa, Denmark; 2Institute for Regional Health Research, The University of Southern Denmark, 5000 Odense, Denmark; 3Danish Artritis Hospital, 6400 Sønderborg, Denmark; 4Brain and Nerve Diseases, University Hospital Sønderjylland, 6200 Aabenraa, Denmark; 5Medical Diseases, University Hospital Sønderjylland, 6200 Aabenraa, Denmark

**Keywords:** medication dispensing, medication service, pharmacy technician, patient safety, adverse drug events

## Abstract

Background: This study aims to evaluate medication dispensing by pharmacy technicians at a geriatric inpatient ward at a Danish hospital. Methods: Four pharmacy technicians were trained in delivering a dispensing service at a geriatric ward. At baseline, the ward nurses recorded the time spent dispensing the medication and the number of interruptions. Similar recordings were completed twice during the period in which the pharmacy technicians delivered the dispensing service. Satisfaction among the ward staff with the dispensing service was assessed by a questionnaire. Reported medication errors were collected during the dispensing service period and compared to a similar time period during the previous two years. Results: The time spent on dispensing medications was on average reduced with 1.4 h per day ranging from 4.7 to 3.3 h per day when the pharmacy technicians performed the service. Interruptions during the dispensing process decreased from a daily average of more than 19 times to an average of 2–3 per day. The nursing staff reported positive feedback on the medication dispensing service provided, especially about easing their workload. There was a tendency toward decreased reporting of medication errors. Conclusion: The medication dispensing service performed by the pharmacy technicians reduced time spent on dispensing medication and increased patient safety by reducing interruptions during the process and decreasing the number of medication errors reported.

## 1. Introduction

Medication errors which can lead to adverse drug events are associated with increased morbidity, mortality, and hospital admissions as well as increased cost and decreased patient health-related quality of life [1,2,3,4,5]. Due to the high prevalence and cost burden of medication errors, the World Health Organization set the goal to reduce medication errors by 50% in 2022 [2].

It has been estimated that 32–38% of medication errors that harm patients are attributable to the administration phase, including the dispensing phases [6,7,8]. Some of these medication errors may be prevented by correct dispensing and administration of the prescribed medications [1,2,4,6,7]. Nurses are often the healthcare providers responsible for dispensing and administering medication to hospitalized patients [3,9,10]. Medication errors may be prevented if distractions, interruptions, a stressful environment, and workload are reduced [2,3,4,11]. Interruptions can be broadly categorized as either individual—interruptions by other healthcare professionals, patients, or relatives—or technical, such as missing equipment or alarms [8]. Interruptions are mostly initiated by nurses themselves through face-to-face communication or by answering telephone calls [8,11].

Nurses at an English hospital have provided positive feedback about pharmacy technicians easing their workload owing to the medication round being reduced due to fewer interruptions spent on searching for missing medication [12].

These improvements may be accomplished by introducing pharmacy technicians at wards to perform the task of dispensing medication. Since pharmacy technicians cannot contribute to the nursing care provided by nursing staff, they would subsequently experience fewer interruptions and deliver the service faster, and they would have more time to care for patients.

In Denmark, all healthcare professionals are obligated to report any medication error, which is categorized as an unintended event [13]. Unintended events are classified as a “near event” or “an actual event”, with the “near event” being an event that was stopped before an unintended event happened. The National Coordinating Council for Medication Error Reporting and Prevention have standardized the categorization of medication errors, resulting in the Medication Error Index [14,15]. The index starts with the least severe category, Category A, which corresponds to the Danish “near event”, and ends with Category I, a fatal outcome [15]. In 2020, a total of 11,404 medication errors, including errors in relation to tables, vaccines, and liquids, were reported from Danish hospitals to the national reporting system [16]. The medication errors are categorized according to harm. Most reports from Danish hospitals in 2020 were medication errors which caused “no harm” to the patient, with 7273 reports (63.7%) [17]. The remaining categorizations of medication-related error reports were distributed as follows: 2244 “mild harm” (19.7%), 1111 “moderate harm” (9.7%), 134 “severe harm” (1.2%) and 10 “fatal” (0.1%) [16]. The remaining 632 could not be categorized [16].

In some countries it has been estimated that approximately 6–7% of hospital admissions are medication-related, with over two-thirds of these considered preventable and thus potentially due to errors [1]. Indeed, the risk of experiencing a medication error may be especially high among elderly patients since they are often treated with more medications [18,19].

Therefore, this study aims to evaluate medication dispensing by pharmacy technicians at a geriatric ward.

## 2. Materials and Methods

The study was conducted at the geriatric ward of Sønderjylland Hospital in Denmark, a ward containing 22 beds, 10–14 nurses, and additional nurse assistants during the study period. The study period commenced in March 2021 and ended the following August. The nurses conducted baseline measurements for a week in March. The pharmacy technicians conducted their measurements twice during the dispensing service: one week in April and one week in the beginning of August.

### 2.1. Pharmacy Dispensing Service

The dispensing service was developed and piloted in close collaboration with the ward management; the timeline is presented in Figure 1. Prior to this, the pharmacy technicians (called “pharmaconomists” in Denmark) were trained in delivering the dispensing service.

All tablets and capsules were dispensed by the pharmacy technicians after the ward round at noon to cover the time period of 6 p.m. to 12 a.m. the next day. Four technicians were trained, and they delivered the dispensing service during the pilot study period.

In the morning, a prescription review was performed to allow time for ordering relevant oral medication if not available in the ward. Prescribed medications were substituted to the standard assortment at the ward. Unusual medications were obtained from the patient or brought into the hospital by patient relatives. Administration times were adjusted to 8 a.m. + 12 p.m. + 18 p.m. + 22 p.m. if possible, to allow predictability of the daily routines for the nursing staff. The nursing staff remained responsible for administering the medication to the patients.

### 2.2. Dispensing Time and Interruptions

To assess usual dispensing practices, the nursing staff recorded time spent on dispensing and interruptions during one week in March 2021. The nursing staff dispensed and administered medication to the patients; they were responsible for 3–4 patients during their day shift. During the evening shift, 1–2 nursing staff dispensed medication to all the patients.

Similarly, pharmacy technicians recorded time spent on dispensing and interruptions during one week in April and one week in August 2021. They also recorded the time spent adjusting administration times and the time spent acquiring medication not included in the standard assortment at the ward.

The arrows in Figure 1 present the process, the recording, and the end evaluation of the pilot project.

### 2.3. Medication Errors

When medication errors were reported, the severity of harm was registered by the person who discovered the medication error [13,16]. “No harm” indicates that the medications did not cause the patient any harm, and “mild harm” indicates that the patient experienced small temporary harm, which did not require further treatment or care [16]. “Moderate harm” indicates harm resulting in hospitalization or treatment by a general practitioner. “Severe harm” signifies permanent harm requiring hospitalization, treatment by a general practitioner, increased care, or emergency life-saving treatment. Finally, “fatal” harm comprises mortality [16].This corresponds to the Medication Error Index Category B to D, Category E to F, Category G, Category H, and lastly Category I [15]. All medication errors reported from the ward during March–September 2021 were compared to similar time periods in 2020 and 2019. The severity of the reported medication errors was also assessed.

### 2.4. Questionnaire for Nursing Staff

To assess nursing staff satisfaction with the dispensing service, a questionnaire was developed and distributed to all the nursing staff. Due to the study design, pilot study, the questionnaire was not validated. The questionnaire contained 9 statements addressing perceived time reduction in daily care, reduction in medication care complexity, and patient safety. The response categories for each statement were: “not at all”, “to a little extent”, “to some extent”, “to a high extent” and “to a very high extent”. The scale was coded with numbers, where “not at all” = 1, “to some extent” = 2…, and “to a very high extent” = 5. For every statement an average was presented.

## 3. Results

### 3.1. Time Used at Dispensing Medication

In total, the nurses spent 14.6 min per patient per day dispensing at baseline, while the pharmacy technicians spent 11.6 and 10.9 min per patient per day, respectively. The pharmacy technicians used appr. 1.4 h less per day during the dispensing process presented in Table 1. Despite the 1.4 h reduction in time spent on dispensing medication, the pharmacy technicians still managed to dispense the planned medication for the next 24 h to the ward’s patients. The nurses managed to dispense medication corresponding to two patients each while the pharmacy technicians dispensed medication for 8.2 and 9.4 as many patients and still reduced the times used for this process.

When the pharmacy technicians conducted the prescription reviews, they also identified potential suboptimal medication treatments, which were solved in collaboration with the ward staff with the aim of increasing the quality of the individual medication treatment.

### 3.2. Interruptions While Dispensing Medication

These results showed that the nurses were interrupted more frequently compared to the pharmacy technicians during the mentally demanding task of dispensing medication to the patients presented in Table 1.

The nurses were interrupted 97 times during the baseline period, which was 6.5 and 7.5 times higher than the pharmacy technicians presented in Table 1. This equates to the nurses being interrupted when dispensing for all the patients. The pharmacy technicians were interrupted for every fifth to seventh patient. Frequently a nurse would be interrupted during medication dispensing in the medicine room by a telephone call. Other examples were interruptions from nursing colleagues who needed help mobilizing a patient or who needed to deliver a message regarding a patient.

### 3.3. Reports of Medication Errors and Severity of Harm Caused by Medication Error

An increase in the reporting of drug-related medication errors as well as the severity of harm caused to patients was evident from 2019 to 2020, as presented in Figure 2. This increase can be attributed to increased focus on reporting as well as a presumed increased incidence of medication errors, according to feedback from the ward. The pharmacy technicians started dispensing in 2021, after which in general there is a tendency toward fewer reports of medication errors. The current study showed a decrease in reports in relation to incorrect dispensing from 2020 to 2021 after initiating the dispensing service.

Moreover, fewer harmful medication errors were reported in 2021 compared to 2020. From 2020 to 2021 there was a decrease by four times in reports about medication errors without harm to the patient.

### 3.4. Questionnaire for the Nursing Staff

Of the nursing staff at the ward, 18 responded: 10 nurses and 8 assistants, including a care assistant student. The results from the questionnaire showed high satisfaction with the pharmacy technician services among the nursing staff presented in Table 2. High satisfaction was reported especially for issues regarding decreasing the workload of the nursing staff (statements 5 and 6). Additionally, this led to the nursing staff spending more time delivering nurse-specific tasks (statement 9). The nursing staff also reported experiencing fewer medication errors in relation to dispensing. This was further supported by the nursing staff reporting increased patient safety as a result of the dispensing service (statement 4).

## 4. Discussion

In this study, the dispensing of medication by the pharmacy technicians led to a 1.4-h reduction in the time used in dispensing and reduced the number of interruptions. The pharmacy technicians became an integrated part of the ward, and the medication dispensing process became more coherent. Moreover, the nursing staff was satisfied with the dispensing service. Finally, there was a tendency toward a reduction in reported medication errors, a fact acknowledged by the nursing staff.

Optimizing the work processes regarding medication dispensing included simplifying the dosing times to 4 times daily if at all possible as well as simplifying the obtaining and dispensing of medication. Reducing the complexity of the medication treatment reduced the time spent on dispensing in addition to reducing the risk of interruptions. Interruptions during the process of dispensing the medication lead to an increased risk of medication errors [2,3,4,9]. When several patient activities occur simultaneously, nurses have to prioritize their tasks and are involved in complex decision making, hence an increased risk of medication errors, especially if the nurses attempt to solve several tasks simultaneously [4,8].

The current study showed that nurses were interrupted more frequently than pharmacy technicians. The same interruption pattern was found by Elganzouri et al. [5], where nurses were interrupted or distracted for every individual patient for whom they dispensed medication [5]. In the literature, nurses reported different kinds of interruptions that can occur during medication administration and dispensing [8,11]. Palese et al. [8] found that the most frequent interruption was obtaining medication or materials not available on the medication trolley during the preparation phase [8]. In the present study, pharmacy technicians obtained missing medication; hence, this would not become an interruption for the nurses when they needed to dispense additional medication to the patient. In the present study, the tendency toward fewer medication errors as a consequence of pharmacy technician dispensing could possibly be attributed to fewer interruptions during the dispensing process. Pharmacy technicians are not trained to complete direct patient care tasks, allowing the pharmacy technician to keep their focus on dispensing medication correctly and time-efficiently.

The fewer interruptions experienced by the pharmacy technicians contributed to less time spent on dispensing. Indeed, the 14.6 min spent by nurses when dispensing is comparable to observations by Keohane et al. [10] and observations in the Medication Administration Study, where nurses spent 15 min per instance of medication dispensing including preparation, retrieval, administration (to the patient), and documentation [5,10]. In addition, when the pharmacy technicians dispensed medicine for the following 24 h, the dispensing process was optimized, saved time, and became more coherent. This freed up time for the nurses to carry out patient care tasks, which was supported by results from the questionnaire. These findings are supported by Seston et al. [12], who also reported reduction in time spent by the nurses, fewer interruptions during the dispensing process, reduction in omitted doses, and satisfaction with pharmacy technicians providing medication dispensing services [12].

The pharmacist technicians became an integrated part of the ward. The geriatric ward treated its patients with a multidisciplinary approach in general, using competencies provided by physicians, nurses, care staff, physiotherapists, ergotherapists, etc.; hence, the pharmacy technicians were smoothly integrated as a part of the daily routines, especially by providing medication information.

We did not measure the quality of the medication treatment, but it is likely that the pharmacy technicians contributed with additional quality due to their prescription review. This includes the substitution of medications to the standard assortment used in the hospital, action on potential suboptimal treatment such as duplicate prescriptions, odd administration times, non-recommended dosing schedules, etc., all of which were discussed with the ward staff.

We did find a tendency toward reduction in medication errors. This is consistent with reports in the literature of a higher risk of medication errors when nurses are stressed and interrupted during the medication handling process [2,3,4]. For instance, dose omission is an example of medication errors, which are required to be reported to the Danish authorities [16,19]. Dose omissions may be prevented if they are due to, e.g., lack of medication at the ward’s storage or a complex dosing regimen. The pharmacy technicians in the current study did address both these types of omissions by providing assistance to obtain medication and reduce the complexity of the dosing regimen.

Finally, there is an acute shortage of nurses in Denmark; therefore, it may be helpful to use pharmacy technicians to dispense and possibly administer medication for patients to assist patient care and possibly improve patient safety.

## 5. Conclusions

In conclusion, we found that the medication dispensing service performed by the pharmacy technicians improved working processes, and additional time was freed up for the nursing staff for direct patient care, improving patient safety and without compromising medication treatment quality. The medication dispensing service performed by the pharmacy technicians has been adopted at the geriatric ward. Future studies are needed to elaborate on the effect of patient safety according to reports of medication errors.

## Figures and Tables

**Figure 1 pharmacy-11-00082-f001:**
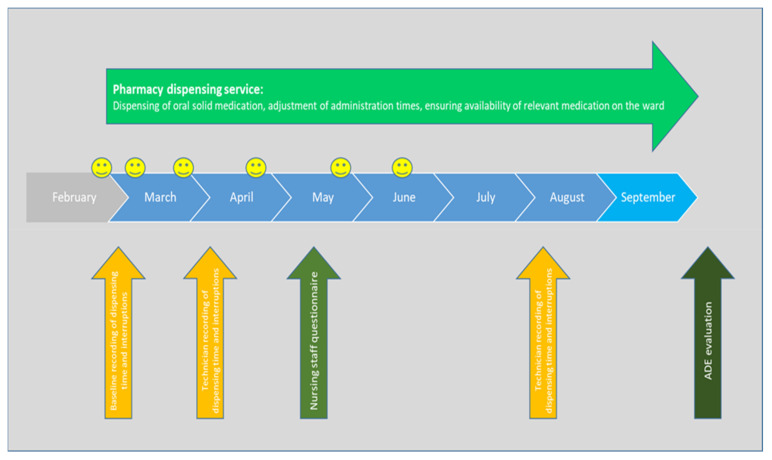
Process and evaluation of the pilot project. Smileys indicate progress meetings with pharmacy staff and ward management.

**Figure 2 pharmacy-11-00082-f002:**
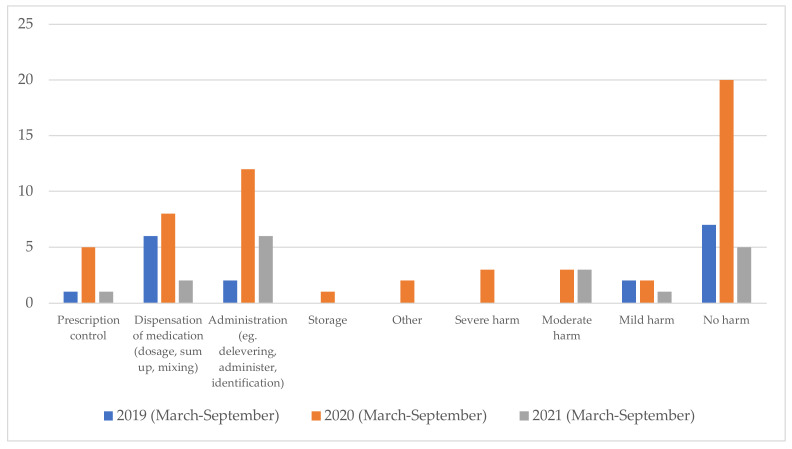
The number of categorized drug-related medication errors and the number of categorized reports about the harm the medication error has caused the patient reported at the geriatric ward in a Danish hospital. The reports are from the geriatric ward from March to September; blue represents 2019, orange 2020, and grey 2021.

**Table 1 pharmacy-11-00082-t001:** Instances of dispensing time used in dispensing medication per day for nurses and pharmacy technicians and the interruptions during the dispensing of medication (Ptt = Patient).

	Nurse (Baseline) (March 2021)	Pharmacy Technician (April 2021)	Pharmacy Technician (August 2021)
Avg. number of dispensing instances per day	38.4 (≈2 disp. per ptt)	16.4 (=number of ptt)	18.8 (=number of ptt)
Prescription review time per day incl. adjustment of time of administration	0 min	34 min	30 min
Clarification of missing medication to be obtained ^1^	.	10 min	12 min
Times used at obtaining and verifying medication per day ^2^	60 min	16 min	20 min
Dispensing time per day	220.8 min (3.7 h)	130 min (2.2 h)	142 min (2.4 h)
Total time per day (3)	280.8 min (4.7 h)	190 min (3.2 h)	204 min (3.4 h)
Number of interruptions per week	97	15	13
Avg. number of interruptions per day	19.4	3.0	2.6
Number of interruptions per dispensation per day	0.5 (i.e., every 2nd disp. ≈ to appr. every ptt)	0.2 (i.e., every 5th ptt)	0.14 (i.e., every 7th ptt

^1^ Estimated based on data collection from August 2021 12 min/18.8 patient times 16.4 patients. ^2^ Estimated based on questionnaire: approx. 30 min per nursing staff in a week.

**Table 2 pharmacy-11-00082-t002:** The statement the nursing staff was presented in the questionnaire and the avg. score. The scale was coded using numbers, with “not at all” = 1, “to some extent” = 2…, and “to a very high extent” = 5. The number in the (X) indicates the responses.

Statement	Average	Statement	Average
1. Do you think that the pharmacy has become an integrated part of daily work in the ward?	4.1 (18)	2. Have *fewer dosing* times given more peace of mind in the workflow at the ward?	4.2 (18)
3. Do you feel safe in administering medicine to patients, even if you have not dispensed it yourself?	3.9 (18)	4. Do you think the pharmacy’s work has improved patient safety in the ward?	3.7 (18)
5. Has the workload in the morning been reduced after the medicine is dispensed by the pharmacy?	4.5 (18)	6. Has the workload generally (not only in the morning) been reduced after the medicine is dispensed by the pharmacy?	4.1 (18)
7. Have you experienced medication errors in relation to the dispensing of the medicine, while the pharmacy has conducted the dispensing?	1.4 (18)	8. Do you spend less time to obtaining medicine from other wards?	4.1 (18)
9. Do you have a better opportunity to spend time with the patients after the medicine is dispensed by the pharmacy?	4.1 (17)		

## Data Availability

The data supporting the results are included in the article; hence, no further data are available.

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
