# Peer review of "Medication Dispensing by Pharmacy Technicians Improves Efficiency and Patient Safety at a Geriatric Ward at a Danish Hospital: A Pilot Study"

_pharmacy, 2023, doi:10.3390/pharmacy11030082_

Round 1

Reviewer 1 Report

The authors present an interesting pilot study that merits publication in Pharmacy. I have a number of points to be considered by the authors before the manuscript may be accepted.

Title: techniciansa geriatric ward, a pilot study. Add (after geriatric ward): in a Danish hospital. Also consider to add something about efficiency. The study shows two major results: one is efficiency, time reduction if dispensing is done by pharmacy technicians, and the other one is a reduction of medication errors, thus increased patient safety.

I think the authors should chose for a fixed sequence, first efficiency and second safety, or the other way around, and structure the entire manuscript consequently along the chosen line / sequence. Now, in the abstract first focus is on efficiency and safety comes next, while the introduction starts with safety. Et cetera for the rest of the manuscript (Results for instance starts with efficiency).

Abstract: also add that the study has been done at a geriatric ward in a Danish hospital.

Abstract: from 4.7 to 3.3 hours (4.7-3.3 hours is not so clear).

Introduction: the paragraph covering lines 52-54 is about efficiency only. The text above and below is about medication errors. I suggest to combine everything about medication errors. See also my comment on the sequence of topics.

Introduction: the aim is too limited as it only mentions medication errors.

Which handlings were included in the dispensing by pharmacy technicians? Did it also include a control of the medication in relation to the patients’ records? Were also syringes filled, dilutions made (if necessary)? Where did the work of the pharmacy technicians stop and was taken over by the nurses at the ward?

By whom were medication errors reported? Can you give examples of medication errors that were prevented in the novel setting?

Methods: indicate what the study period was. Also in relation to Methods 2.4 this is essential to add.

Methods 2.1: describe more clearly what the dispensing tasks were. Was it only for oral medication (apparently put in a box per patient per day?) or also (incidental) parenteral medication? Did the pharmacy technicians also help or gave advise how to deal with patients with swallowing problems? Think of crushing (or not) of tablets.

Methods 2.3: was the developed questionnaire validated before it was used?

Results 3.2: what kind of interruptions? Give some examples.

Table 2: add the number of answers on which the average is based.

Figure 2: remove heading from figure. Extend legend if necessary. Why is Geriatric Ward with capitals and nowhere else in the manuscript? What is moderate harm? How determined?

What I miss is a statistical comparison of outcomes, where appropriate. For instance, time for dispensing in the old situation versus in the new situation, numbers of errors. Now only means/averages are presented.

Conclusion: clear, but will there be a follow-up? The results are positive and will this result in an adaptation of the activities at the ward? Thus, in the future pharmacy assistants for dispensing and no longer nurses? Or will there be a next study first, as the current study is a pilot?

Please check the manuscript carefully for typo’s and linguistic errors; I found several (e.g., line 17: by aquestionnaire; line 17: reported medication errors were collected; heading 3.1: time used to; line 131: this is equivalent to; line 137-138 is crooked; line 179: the current study shows.

See above.

Reviewer 2 Report

This is a carefully conducted study and of subsantial value at a time when medical professionals are learning how to work better in teams. 

My comments are directed to your study having wider impact and readership.

Introduction, Methods

There is a large literature on medicaton errors (as distinct from Adverse drug events) which you do not cite. Errors include transcription, wrong medication, wrong patient, dosage, frequency and route of administration errors. To place your study within the literature please use an index to categorise the different errors, such as the National Coordinating Council for Medication Error Reporting and Prevention (NCC MERP Index).

You make an important contribution on counting interruptions.  

Emergency physicians and nurses especially can have problems focusing  when constantly interrupted.There are also studies of how many individuals (staff, visitors) enter patients' rooms per day what assessments and changes they make and how long they stay there. Can you please research and cite this literature to compare it with your findings?

Excellent.

Round 2

Reviewer 2 Report

Thanks to the authors for their changes to this excellent study.